# A Focal Adhesion Filament Cross-correlation Kit for fast, automated segmentation and correlation of focal adhesions and actin stress fibers in cells

Lara Hauke[1], Shwetha Narasimhan[2], Andreas Primeßnig[1], Irina Kaverina[2]*, Florian Rehfeldt[1,3]*

**1** Third Institute of Physics—Biophysics, Georg-August-University Göttingen, Göttingen, Germany, **2** Department of Cell and Developmental Biology, Vanderbilt University, Nashville, TN, United States of America, **3** Experimental Physics I, University of Bayreuth, Bayreuth, Germany

☯ These authors contributed equally to this work.
* irina.kaverina@vanderbilt.edu (IK); florian.rehfeldt@uni-bayreuth.de (FR)

**Data Availability Statement:** All microscopy images, the processed and annotated images and the analysis files and tables are publicly available

## Abstract

Focal adhesions (FAs) and associated actin stress fibers (SFs) form a complex mechanical system that mediates bidirectional interactions between cells and their environment. This linked network is essential for mechanosensing, force production and force transduction, thus directly governing cellular processes like polarization, migration and extracellular matrix remodeling. We introduce a tool for fast and robust coupled analysis of both FAs and SFs named the Focal Adhesion Filament Cross-correlation Kit (FAFCK). Our software can detect and record location, axes lengths, area, orientation, and aspect ratio of focal adhesion structures as well as the location, length, width and orientation of actin stress fibers. This enables users to automate analysis of the correlation of FAs and SFs and study the stress fiber system in a higher degree, pivotal to accurately evaluate transmission of mechanocellular forces between a cell and its surroundings. The FAFCK is particularly suited for unbiased and systematic quantitative analysis of FAs and SFs necessary for novel approaches of traction force microscopy that uses the additional data from the cellular side to calculate the stress distribution in the substrate. For validation and comparison with other tools, we provide datasets of cells of varying quality that are labelled by a human expert. Datasets and FAFCK are freely available as open source under the GNU General Public License.

## Introduction

The shape and mechanics of biological cells depends largely on the cytoskeleton, a dynamic network that functions as the cellular cytoskeleton and produces contractile forces acting on their environment, such as the extracellular matrix (ECM) or neighboring cells. A predominant and essential part of this network is made up of actin filaments that act as structural

online: https://doi.org/10.5281/zenodo.5082933 www.filament-sensor.de.

**Funding:** I.K.; R35-GM127098; The National Institute of General Medical Sciences (NIGMS) of the National Institutes of Health (NIH); https://www.nigms.nih.gov/ I.K.; R01-DK106228; The National Institute of Diabetes and Digestive and Kidney Diseases (NIDDK) of the National Institutes of Health (NIH); https://www.niddk.nih.gov/ S.N.; 18PRE33990479; https://www.heart.org/; the American Heart Association (AHA); https://www.heart.org/ F.R. SFB 755 - B08, Deutsche Forschungsgemeinschaft (DFG), www.dfg.de F.R. SFB 937 - A13, Deutsche Forschungsgemeinschaft (DFG), www.dfg.de F.R. Open Access Publishing funding programme of the University of Bayreuth and the German Research Foundation (DFG), https://www.ub.uni-bayreuth.de/en/index.html, www.dfg.de The funders had no role in study design, data collection and analysis, decision to publish, or preparation of the manuscript.

**Competing interests:** The authors have declared that no competing interests exist.

elements and, importantly, are capable of producing contractile forces when co-assembled with myosin II mini-filaments into contractile stress fibers [1].

Geometry and rearrangement of stress fibers is a critical factor during mechanical interactions between the cell and the ECM in many processes (e.g. adhesion, migration, etc.) and must be quantitatively assessed to elucidate the complex mechanical interplay of cells with their surroundings. Interestingly, the pattern of stress fiber formation in human mesenchymal stem cells reveals an optimal matrix elasticity $E$ yielding an anisotropic and polarized acto-myosin fiber structure, which functions as an early morphological marker of mechano-guided differentiation [2, 3]. This requires a quantitative analysis of the filament structure (e.g. by a simplified order parameter $S$ known from liquid crystal theory as introduced earlier [2], that builds on the unbiased and automated segmentation of stress fibers. Various approaches exist to address this task, among which our recently developed FilamentSensor analysis tool allows for automated detection and quantification of stress fiber structures [4]. However, for a complete functional analysis of cell and matrix mechanics, quantification of both stress fibers and their associated focal adhesions is needed.

Cells adhere to the ECM or surrounding cells via cell-matrix and cell-cell contacts, respectively. These structures function as biochemical anchors and are key to the signaling and mechanical interactions of cells with their surroundings. Focal Adhesions (FAs) are cell-matrix anchors based on trans-membrane proteins integrins, with a multitude of associated proteins on the cytosolic side. Serving as the interface between the SFs and the ECM, FAs have several functions, such as providing cellular attachment to the substrate, transducing contractile forces to the ECM and facilitating bi-directional transmembrane signaling [5]. At the cytosolic side, FAs are structurally and dynamically linked to the ends of SFs (see Fig 1A–1C). The formation and maturation of FAs is dependent on actomyosin-generated tensile forces applied on them through associated SFs [5]. In turn, signaling pathways that are mechanically triggered at adhesions lead to actin polymerization and elongation of the fibers at their FA-associated termini [6]. Thus, there is an intricate, dynamic association between FAs and SFs that needs to be quantified to fully elucidate their cellular functionality.

Cellular SFs are broadly classified as transverse arcs, dorsal SFs and ventral SFs based on their FA association, which underlies their varied roles (Fig 1D) [1]. Actin transverse arcs, which are not associated with FA but rather embedded into the cortical actin meshwork at their termini, are contractile structures that contribute to cell shape but not do not directly exert force onto the environment. Dorsal SFs are associated with FAs at one end and with transverse arcs on the other end. Although they are non-contractile due to their negligible myosin II content, they can exert forces on their terminal adhesion through their association with transverse arcs. Ventral SFs, which are connected to FAs at both ends, are contractile structures that generate majority of cellular traction forces on the substrate [7, 8]. Due to this natural linkage of SFs and FAs, cytoskeletal studies often result in cells with an observed actin SF phenotype having an associated FA phenotype [9–13]. Therefore, incorporating detection of SF-FA coupling in studies would greatly facilitate the complete analysis of their structure and function in cells.

Manual evaluation and analysis of FAs and SFs is a laborious, time-intensive process and is always at risk due to the observer's bias. Recently, this process has been aided by several automated analysis tools and algorithms that are optimized either for focal adhesion analysis (such as the Focal Adhesion Analysis Server [14], PAASTA [15], or Buskermolen's segmentation algorithm [16]), or stress fiber analysis (such as previous version of FilamentSensor [4], Cyto-Seg [17], SFEX [18], and MatLab scripts like Rogge's FSegment [19]).

However, a tool for speedy, unbiased quantification of SFs, FAs, and their mutual coupling is yet missing. Here, we present an integrated FA-SF analysis module called the Focal

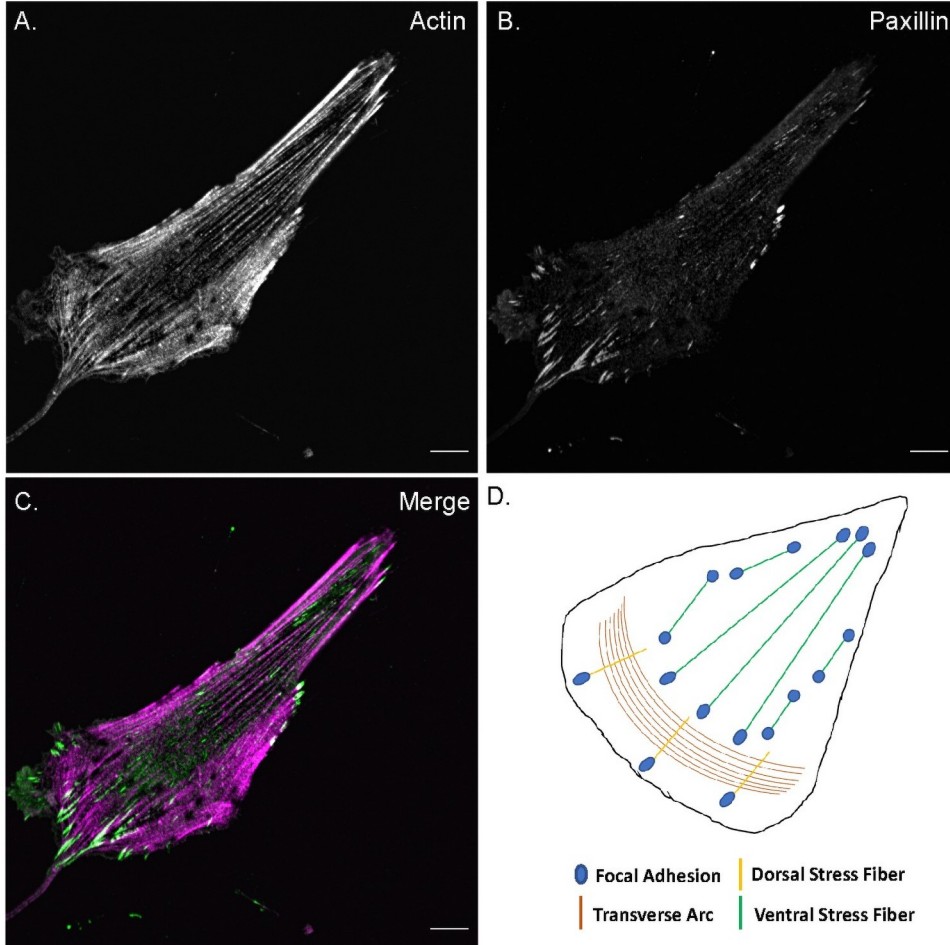

**Fig 1. Stress fibers and focal adhesions.** Confocal fluorescence microscopy images of an MRC5 cell stained for **A)** actin filaments (phalloidin) and **B)** focal adhesions (paxillin). **C)** Merged color image of the cell with actin filaments in magenta and adhesions in green. All images are of the ventral plane of the cell, scale bar—10 μm. **D)** Schematic illustration of different stress fiber subtypes and their association with focal adhesions.

Adhesion Filament Cross-correlation Kit (FAFCK). This tool is based on our previously published FilamentSensor analysis tool, with added capacities for adhesion detection and characterization, filament analysis and coupled FA-SF correlation for stacks or pseudo-stacks of images with similar properties to streamline analysis of huge datasets. FAFCK detects and quantifies FAs and SFs by means of location, area, length, width, aspect ratio and orientation, with capacity for exporting this information enumerated for each frame, allowing for comprehensive further data analysis (e.g. Python, Matlab, etc.) to elucidate cell and matrix mechanics. Our software package will be particularly helpful for sophisticated mechanical measurements and analysis such as model based traction force microscopy (MBTFM) experiments [7] that takes advantage of the *a priori* determined positions of focal adhesions and stress fibers in addition to the displacement field in the substrate.

## Results

The Focal Adhesion Filament Cross-correlation Kit (FAFCK) is a comprehensive FA-SF analysis software consisting of two modules: the FASensor, for adhesion detection and the

FilamentSensor, for actin filament detection, both of which connect through a correlation function for paired characterization of these structures. To correlate an adhesion with the associated actin filament in the cell, the software relates each adhesion object detected by the FASensor module with corresponding filaments that are detected by the FilamentSensor module. As stand-alone programs with a shared GUI, both routines can be used independently as well.

## Segmentation of focal adhesions by FASensor

The FASensor is the adhesion detection module in the software. It is a robust tool for detection of point-like structures partly based on the FilamentSensor [4]. Based on adapted ImageJ routines (Fig 2), it analyzes the adhesions in an image as objects which can be exported with characteristics and IDs, with multiple customization options to improve accuracy as desired by the user.

Adhesion detection analyzes and segments the input image (usually a grayscale immunofluorescence (IF) micrograph) of focal adhesions. The module is split into Main, Pre-processing and Focal Adhesion output sections. All images are shown in panels on the right- including the original image of adhesions, the pre-processed image, the thresholded image, and the image with overlay of filaments detected from the filament input (Stress Fiber Overlay). The windows can be split from the interface and zoomed in for user ease. The pre-processing tab allows the user to add optional filters to the image in order to improve the signal to noise ratio and normalize the image. Filters included are the Gauss filter, Laplace filter, Line Gauss filter, Cross-correlation filter, and Enhance contrast filter. Filter queues can be saved for reuse. The main tab has thresholding controls with automated protocols. The levels can also be altered manually to produce the desired binarized image. Additional filters are provided for defining the minimum or maximum pixel number per adhesion and the maximum amount of clusters allowed in one image. Fig 3 shows an examplary view of the main window of the software with focal adhesions and stress fibers detected but correlation not yet run. Detailed explanation of the submenus and functions can be found in the tutorial provided with the software.

On clicking 'Process Focal Adhesions', the adhesion objects are detected. For each adhesion detected, the outline is derived, and a convex hull is calculated. The main axis is set for the points farthest away on the convex hull and for the points farthest away from the main axis, the side axis is set. The aspect ratio, orientation, and center for each focal adhesion is also calculated. The module also allows for further close customization of the detected objects by the user to obtain the most accurate result. In cases where nearby adhesions have been detected as a single one due to poor signal-to-noise ratio, overlap, artifacts, etc. the user has the option to draw a line on the thresholded image and separate the adhesions at their discretion (see Fig 4A). Once the lines have been drawn to separate all adhesions as desired, the adhesions can be re-processed to get the split objects in a new map.

The detected FAs are displayed in the table in the Focal Adhesion tab. The ID, XY center position, Length of main axis, Length of side axis, Angle, Area, and Area ellipse of each adhesion are listed in the table. The user can choose to discard a detected adhesion object by selecting the object in the Focal Adhesion original window, on which the boundary turns red, and clicking the remove button under the table. This allows the user to closely edit the adhesion map obtained from the software to remove any inconsistencies based on their expertise. The output focal adhesion map can be exported as a binary mask with outlines and optional numbering with IDs. The output table can be exported as a 'CSV' file and the adhesion detection can be exported as a project 'XML' file.

We illustrate the usage of FASensor with the input image of an MRC5 human fibroblast cell showing adhesions (paxillin) (see Fig 4B). This input file was preprocessed using Laplace and

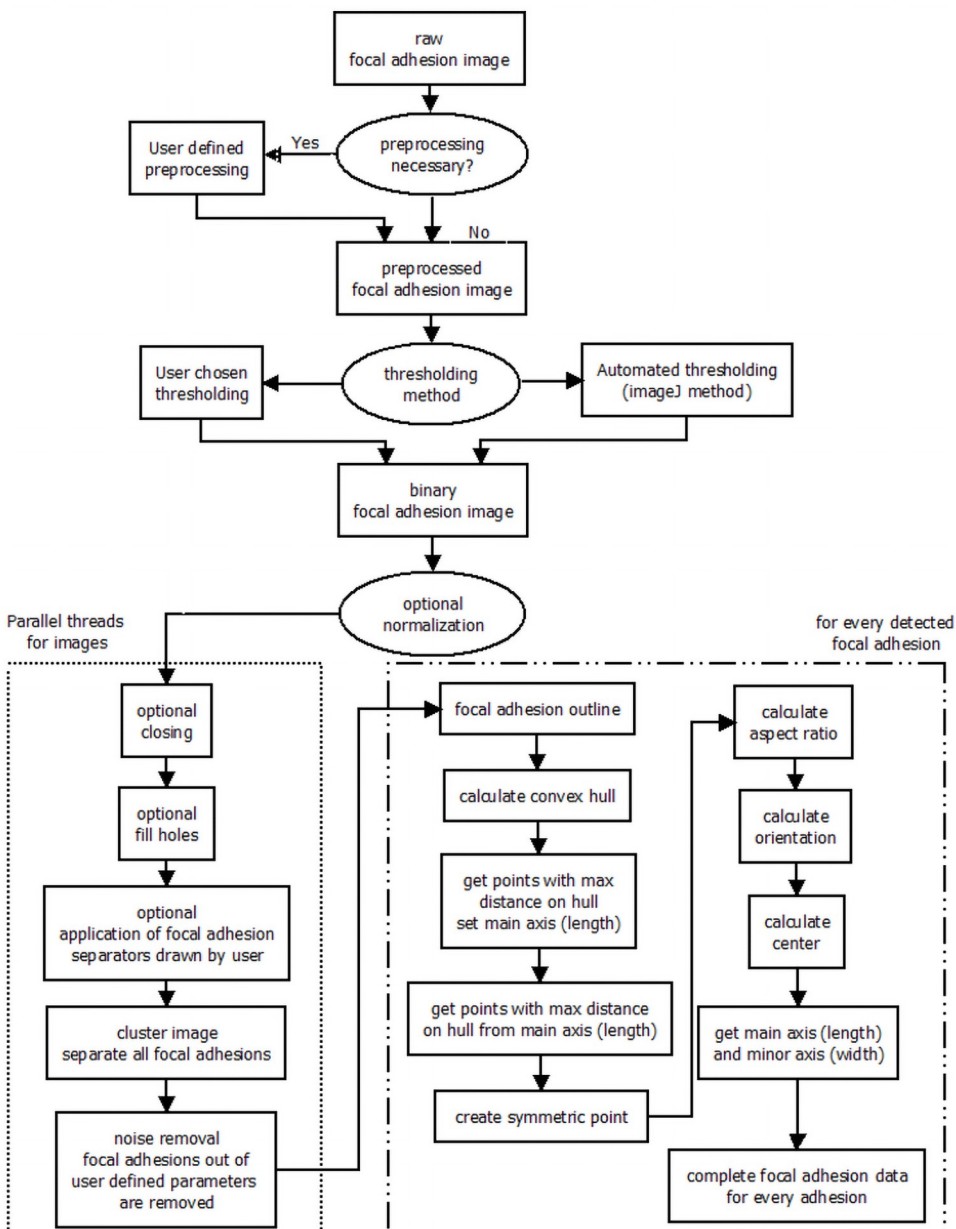

**Fig 2. Workflow of adhesion detection by FASensor module.**

Gauss filters and thesholded using Intermodes algorithm. With a minimum limit of 10 pixels per adhesion, which corresponds to 0.144 $\mu m^2$, adhesion objects were detected by the software (Fig 4C).

In the post-thresholding section, there is the option to add or opt-out of the closing and filling holes algorithms, by which seemingly disparate objects can be detected as one, especially in cases of large, single adhesion plaques whose signal is not uniform (Fig 4D). A large, boundary adhesion plaque that is detected as split pieces without the closing and fill holes algorithm can be reprocessed with this algorithm in order to assign it as a single object. After the reprocessing step, the user is able to confirm that the newly joined adhesion matches with expectations.

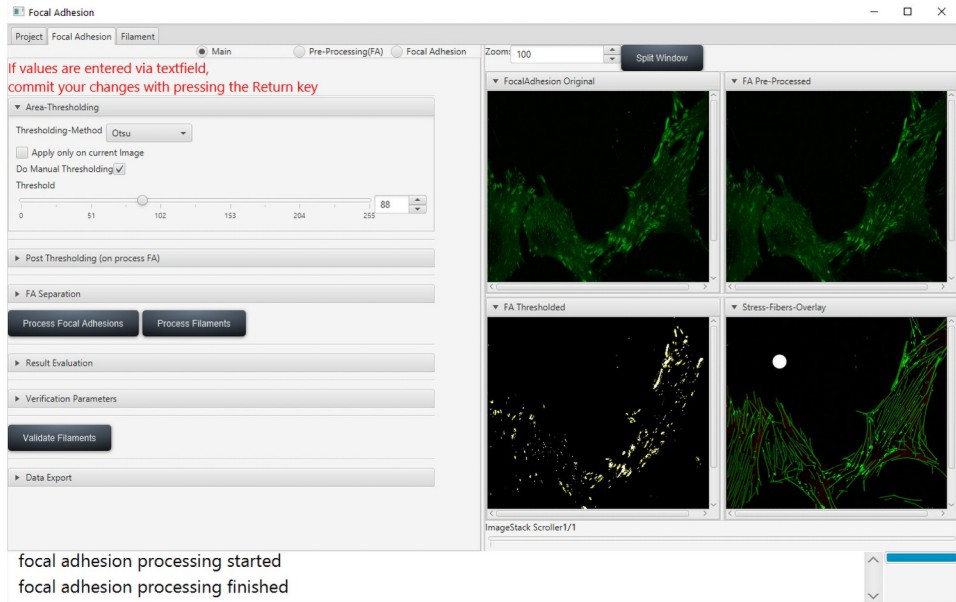

**Fig 3. Main window of the FAFCK software.** Main window of the GUI consisting of several sub-menus at the left to set threshold, FA size restrictions, validation preferences as well as routine for evaluation against external binary masks and data output options. On the right, processing output is shown and can be customized by the user or detached from the main GUI for a larger view.

## Evaluation of the FAFCK output with user generated output

For those using this module to analyze cellular adhesions or stress fibers, it is important to understand how the results compare to their expert opinion and any pre-established routines they already use. To accurately assess the differences between the user's usual routine and the FAFCK output in adhesion detection, our module offers an evaluation option.

In the evaluation panel, a binary adhesion or fiber map generated by the user can be compared with the respective object output generated by FAFCK software (S1 Fig). Before comparison, additional preprocessing can be applied, for example thickening of outlines. The comparison is done in two ways—an objectwise fashion, where from both the user mask and software output, objects are generated and overlap is checked, and in a pixelwise fashion, where each pixel of user mask and software output is taken into account. The minimal required overlap for object matching between the user mask and output can be manually set by the user. The 'export results' option provides images of the comparisons and comparison results in a csv file. The results table lists objects that are found in output when compared with the user's mask, objects that are false positives (present only in the software output image, labelled 'eval not matched') and the missed objects that are present only in the user's image (labelled 'truth not matched'). The pixel sizes of all objects are given along with the number of pixels that overlap in the common objects. The output table also gives the cases where the sensor detects multiple objects in output for one object in the mask marked by the expert (multiMatchesOneToN) and cases where the sensor detects one object in the output for multiple objects marked in the mask by the expert (multiMatchesNToOne).

In the example, shown for FAs, the FASensor output is evaluated against the user generated binary adhesion mask (S1B Fig) from the IF image using Fiji software [20]. On the landing page of the graphical user interface, the user can also import two binary masks of various origins to execute the evaluation without running the software to get an output first. As with all

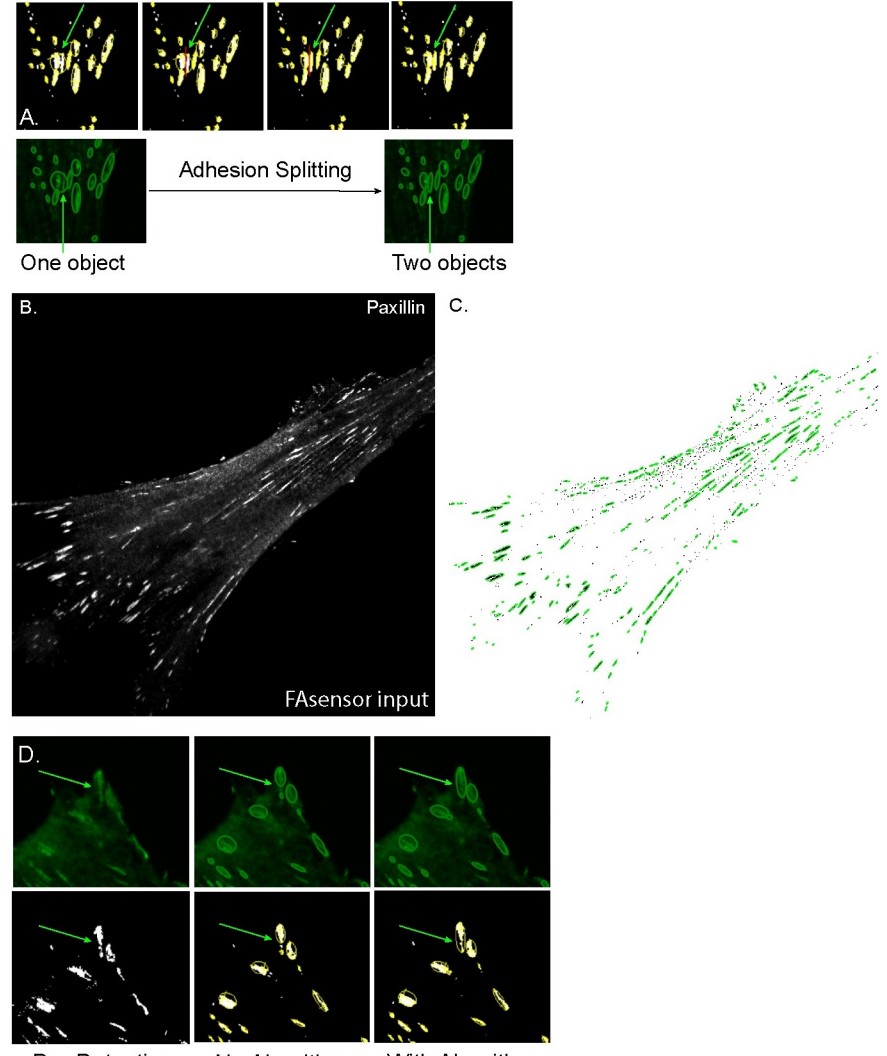

**Fig 4. Segmentation of FAs by FASensor and subsequent optimization. A)** Montage of adhesion splitting capacity of the FASensor module. (Top) Thresholded image of adhesions (white) have objects detected by module (circled by yellow). Red line is drawn by user to split objects where desired. (Bottom) Before and after images of objects detected in the IF adhesion input, that are split. Green arrow indicates splitting objects in Top and Bottom.**B)** Input image of focal adhesions (paxillin) in the ventral plane of an MRC5 cell. **C)** Corresponding segmented adhesion objects (outlined in green) from FASensor. **D)** Montage showing the closing and filling holes function of the module. (Top series) Objects circled by green detected by FASensor from IF adhesion input. (Bottom series) Objects circled by yellow on thresholded image. (Left) Pre-detection by module (Center) Objects detected when closing and filling holes algorithms are not applied. (Right) Objects detected when closing and filling holes algorithms are applied. Green arrow in the Top and Bottom series indicates the adhesion which is detected as multiple objects without the algorithm and detected as a single object with the algorithm.

parts of the software, the evaluation tool can work on OME-TIFF files to provide fast evaluation of large datasets. S1C Fig shows the result maps of objectwise and pixelwise evaluation between FASensor output and the user's mask. The evaluation maps assume the user generated mask as true, highlighting the found and missing categories on it and superimposing false positives from software output on the mask as well. The tables S1D and S1E Fig show the tabulated results for the different categories in objectwise and pixelwise evaluation respectively.

The input routine, in terms of filters and thresholding method used, affects similarity of the FASensor output to user drawn mask. S2 Fig shows how objects detected by FASensor are more similar to the user mask when an appropriate input routine is used instead of default settings. From input of S2A Fig), FASensor output is derived in two ways: unoptimized (filter settings and thresholding that is default coded in the software and not necessarily appropriate for the cell) and optimized (appropriate filter settings and thresholding adjusted by the user). These settings can consist of filters applied to the whole image, manually set thresholding algorithm parameters, applying or not applying the closing and fill holes options, setting boundary conditions for focal adhesion size, and finally separating focal adhesions via user input. Object-wise evaluation of the outputs with user mask (S2B Fig) is shown in S2C Fig for unoptimized output and S2D Fig for optimized output, where found and missing categories as compared to output on the left are highlighted on the user mask and false positives from output are superimposed on the mask as well. In the unoptimized output, the pronounced background signal at the input cell border is fused as large plaques, detected objects deviate from the user mask and many false positives are present. By using an appropriate, optimized input routine, the focal adhesion signal is separated well from the background and adhesions are detected. More detected adhesions match with the user mask and false positives are largely diminished as well. The results are summarized in S2E Fig. There is an increase in the multiMatchesOneToN parameter for the optimized routine, because the optimized input routine finely detects adhesions in the boundary areas of high background, where some of them have been marked as large single adhesions by the user when the signal couldn't be distinguished finely by eye. Thus, several objects detected by the output in these areas are matched to one object marked by the user.

Conversely, if the output had detected a large object from signal that was distinguished as several objects by the user, that would result in an increase in the multiMatchesNToOne parameter.

## FASensor output performance with varying imaging conditions and levels of optimization

To test FASensor's robust detection of focal adhesions on a variety of image qualities, we compiled comparison datasets with varying degrees of blur, in which structures were manually labelled by a human expert for comparison. MRC5 cells immunostained for actin filaments and focal adhesions were imaged on a confocal laser-scanning microscope (S3 Fig) in three conditions with blur introduced in images by altering the size of pinhole to include out-of-focus light. For the Confocal in-focus dataset (Fig 5A), the pinhole size was 1.2 Airy Units (AU), for the Confocal mild blur dataset (Fig 5B), the pinhole size was 3 AU and for the Confocal severe blur dataset (Fig 5C), the pinhole size was 4.7 AU.

Since the FilamentSensor module has been analyzed and published before, we have focused on the FASensor module for manual annotation comparison. We analyzed a set of adhesion images from each imaging condition (in-focus, mild blur and severe blur) in the FASensor software and compared the software results with adhesions manually annotated for the respective images. For manual annotation by the user expert, selected images from the sets were marked for adhesions using the freehand selection tool in Fiji with the aim of being natively user-detected. Images were traced with minimal signal manipulation to compare the base-level manual annotation by eye with the objects traced by FASensor module after processing by software.

To further understand whether and how user involvement such as pre-processing each image in a set differently or splitting ROIs and excluding adhesions makes a significant

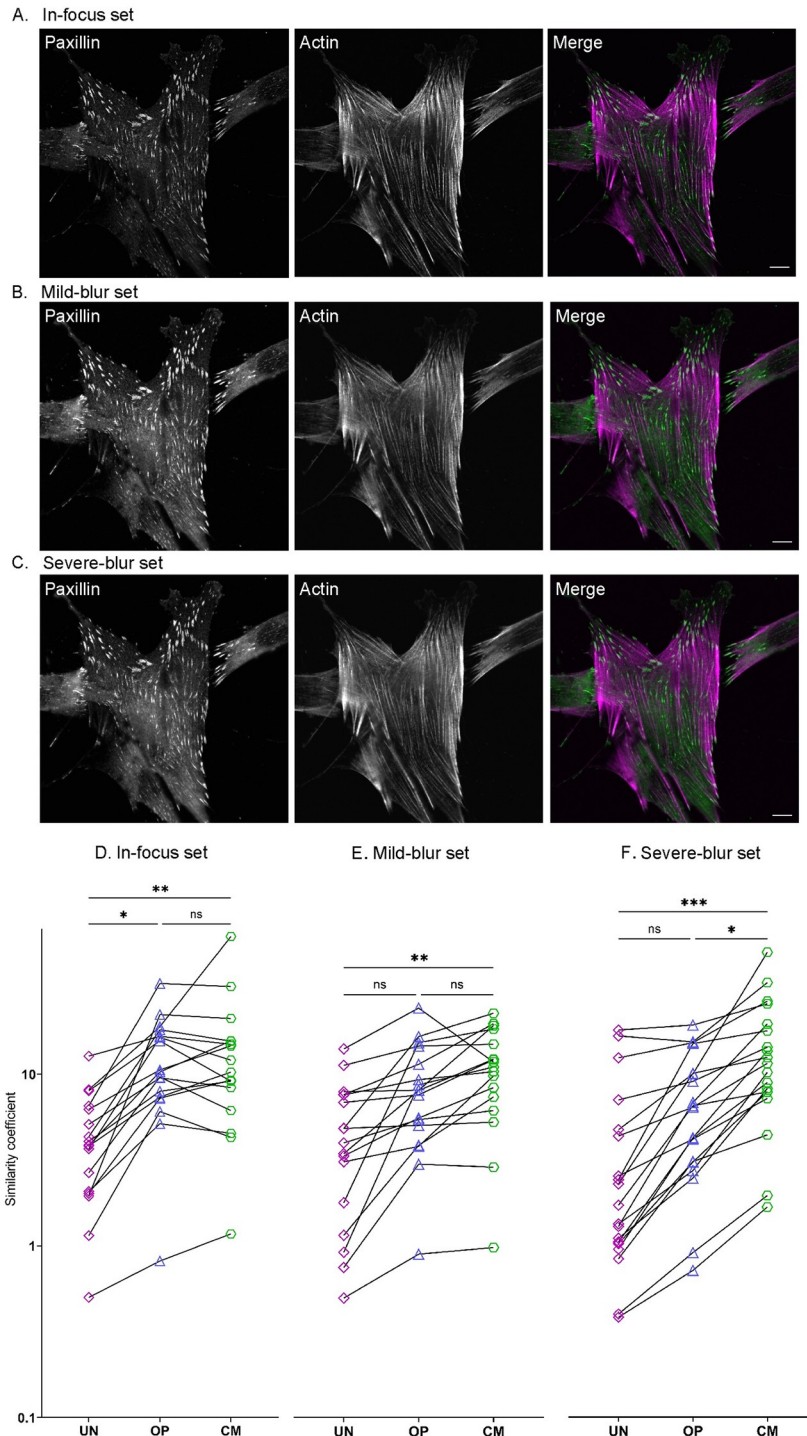

**Fig 5. Datasets' imaging conditions and similarity coefficients ($\mathcal{SC}$) for the datasets by level of optimization. A-C)** Representative images of MRC5 cells fluorescently stained for actin filaments (phalloidin) and adhesions (paxillin) is shown. Scale bar—10 μm. All images were taken with a confocal microscope and are of the ventral plane of the cell. (Left) Grayscale image of the adhesions (Center) Grayscale image of actin filaments (Right) Merged image of the cell with actin filaments in magenta and adhesions in green. A) In-focus setting. B) Mild-blur setting. C) Severe-blur setting. **D)** $\mathcal{SC}$ on the y-axis (logarithmic scale) for unoptimized 'UN' (purple squares), optimized 'OP' (blue triangles) and for customized 'CM' (green hexagons) output of analyzed images of the in-focus set (n = 17, UN $\overline{\mathcal{SC}} = 4.53$, OP $\overline{\mathcal{SC}} = 12.73$ and CM $\overline{\mathcal{SC}} = 14.83$), **E)** Mild-blur set (n = 17, UN $\overline{\mathcal{SC}} = 4.47$, OP $\overline{\mathcal{SC}} = 7.98$ and CM $\overline{\mathcal{SC}} = 10.13$) and **F)** Severe-blur set (n = 19, UN $\overline{\mathcal{SC}} = 3.47$, OP $\overline{\mathcal{SC}} = 5.94$ and CM $\overline{\mathcal{SC}} = 11.74$). *** p <0.001; ** p <0.01; * p <0.05 and ns stands for not significant (p >0.05).

improvement in software results, we used three different optimization levels. In the unoptimized (UN) level, the user sets a single desired input routine with thresholding and pre-processing parameters for all images in the dataset and derives results from the software. There is no optimization for each cell in the dataset and user involvement is low. In the optimized (OP) level, the user sets a custom input routine for each cell with the optimal thresholding and pre-processing parameters and derives results from the software. This optimization uses the software's capability for pre-processing and thresholding to enhance adhesion recognition for every cell according to user's discretion. The user involvement is greater than unoptimized in that every cell has a different optimal setting. In the customized (CM) level, the user sets a custom input routine for each cell and further edits the result by splitting ROIs and deleting adhesions detected so that the result is highly customized and similar to the user manually marking the adhesions. Customization is useful for conditions where the user does not have the time to mark adhesions manually but still desires the detected adhesions to exactly fit to their discretion of the adhesion pattern in an image. The user involvement is thus higher than unoptimized and optimized levels.

Comparison of the software output with the manually marked adhesions gives result categories of adhesions that are found, missed or false positives. To compare these three results in the three optimization conditions, we created a similarity coefficient ($\mathcal{SC}$) for adhesion detection that is as follows:

$$\mathcal{SC} = \frac{\sum Found\ FA\ area}{\sum Missed\ FA\ area + \sum False\ positive\ FA\ area}$$

The higher the coefficient, the more similar the detected adhesions are to the human expert's mask.

For the cells in the In-Focus dataset (Fig 5D), the similarity coefficients show that adhesions detected in OP ($\overline{\mathcal{SC}} = 12.73$) and CM ($\overline{\mathcal{SC}} = 14.83$) sets are significantly more user-similar compared to the UN set ($\overline{\mathcal{SC}} = 4.53$). The similarity coefficient of the OP and CM sets are not significantly different. Just setting optimal pre-processing settings vastly improves similarity of detected adhesions between cells in the in-focus set, even without further time-intensive customization of splitting and deleting detected objects.

For the cells in the mild-blur dataset (Fig 5E), the similarity coefficients show that OP set ($\overline{\mathcal{SC}} = 7.98$) is not significantly more similar than UN ($\overline{\mathcal{SC}} = 4.47$) or CM ($\overline{\mathcal{SC}} = 10.13$), whereas CM is significantly more similar than UN. Thus, in conditions where images have some blur, doing both optimal pre-processing and user customization by splitting and deleting adhesions makes it significantly more accurate.

For cells in the severe-blur dataset (Fig 5F), the similarity coefficients show that OP set ($\overline{\mathcal{SC}} = 5.94$) is not significantly more similar than UN ($\overline{\mathcal{SC}} = 3.47$), but again CM ($\overline{\mathcal{SC}} = 11.74$) is significantly more similar than both other sets. Thus, in conditions where images are blurred, intensive user customization by splitting and deleting adhesions gives the best result.

Aggregate analysis (S3 Fig) of all the adhesions in the sets reveal that false positive adhesions are consistently significantly smaller than found and missed adhesions across optimization levels and missed adhesions are significantly smaller than found adhesions as well. (Fig 6) shows graphs comparing the adhesion objects by area in a set across optimization levels.

Aggregate analysis of the in-focus set (Fig 6A) shows that there is no significant difference between area of the found objects across optimization levels, but both OP (1843) and CM (1802) find more adhesions compared to UN (1451). OP (268) and CM (309) miss less adhesions than UN (660) and CM missed adhesions are significantly smaller than UN.

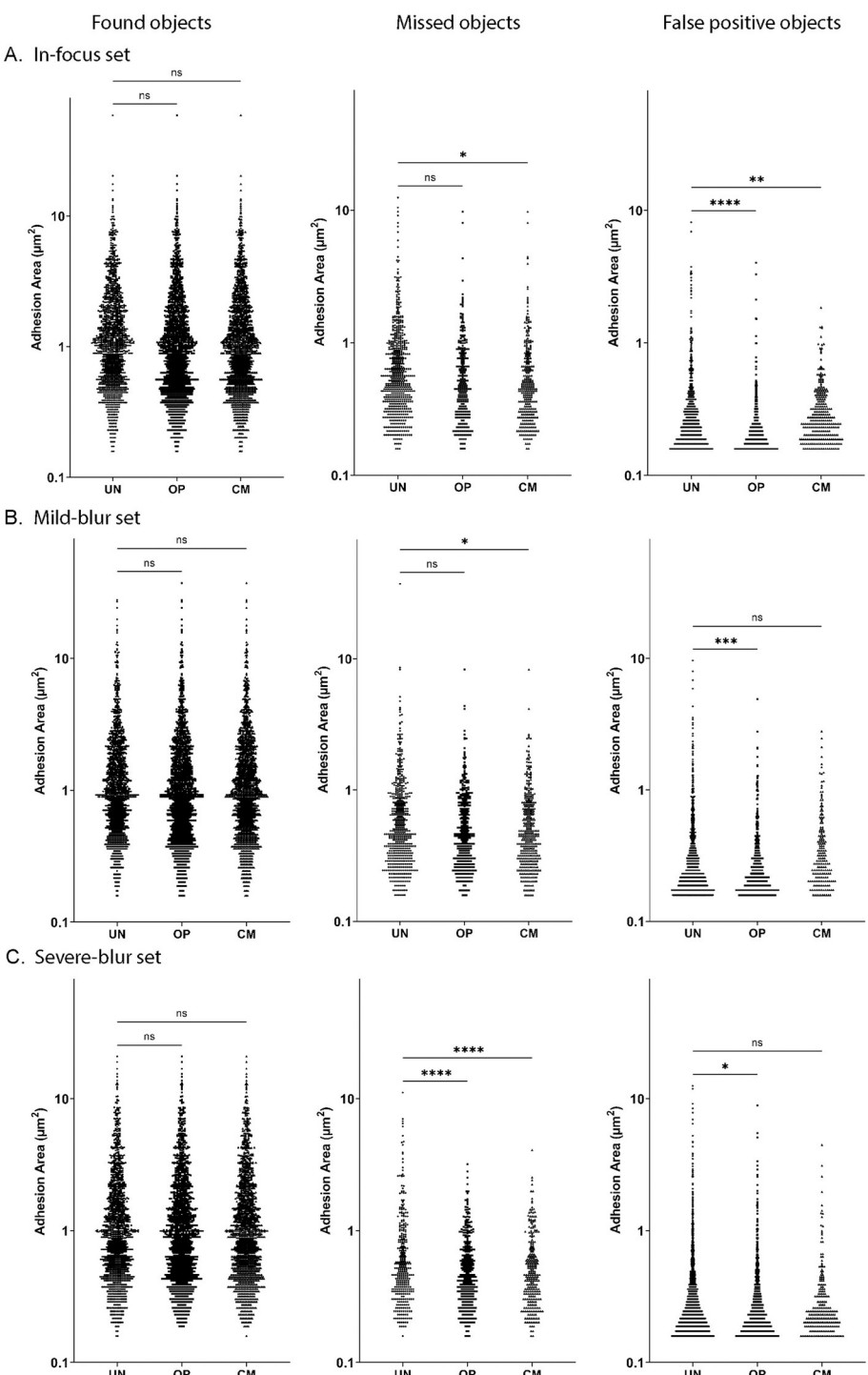

**Fig 6. Comparison of adhesion objects between optimization levels in a set.** Graphs show pooled adhesion objects for un-optimized 'UN', optimized 'OP' and customized 'CM' analysis. Y axis has adhesion area in $\mu m^2$ on a logarithmic scale. Left column shows graphs comparing adhesion objects found in common between user mask and software output. Middle column shows adhesion objects that were missed in output and present only in user mask. Right column shows adhesion objects that are false positive, present only in the software output. **A)** Graphs for in-focus set **B)** Graphs for mild-blur set **C)** Graphs for severe-blur set. **** $p < 0.0001$; *** $p < 0.001$; ** $p < 0.01$; * $p < 0.05$ and ns stands for not significant ($p > 0.05$).

Optimization decreases false positive adhesion area significantly- with CM ($\overline{A} = 0.33$ $\mu$m$^2$) and OP ($\overline{A} = 0.25$ $\mu$m$^2$) adhesions being significantly smaller than UN ($\overline{A} = 0.43$ $\mu$m$^2$) false positive adhesions. OP shows the largest decrease in false positive adhesion size, but CM (296) has the largest decrease in number over UN (497) and OP (465). Thus, optimization of in-focus images primarily increases the accuracy by finding more objects and decreasing false positives by number and area.

Aggregate analysis of the mild blur set (Fig 6B) shows that there is no significant difference between the area of found objects across optimization levels, but both OP (1740) and CM (1700) find more objects compared to UN (1522). In missed objects both OP ($\overline{A} = 0.66$ $\mu$m$^2$) and CM ($\overline{A} = 0.64$ $\mu$m$^2$) have smaller missed adhesion area compared to UN ($\overline{A} = 0.83$ $\mu$m$^2$), and less missed adhesions (OP (361), CM (401)) than UN (579) as well. Comparing the false positive objects, OP has lower $\overline{A}$ ($\overline{A} = 0.31$ $\mu$m$^2$) compared to UN ($\overline{A} = 0.42$ $\mu$m$^2$) and CM ($\overline{A} = 0.43$ $\mu$m$^2$), the number of false positives (560) is lower than UN (791) but higher than CM (188). Thus, in mild blur images, optimization method shows improvement over unoptimized by finding more adhesions and having fewer false positives. Customization shows much fewer false positives compared to other levels.

Aggregate analysis of the severe blur set (Fig 6C) shows that there is no significant difference between area of found objects across optimization levels, but both OP (1520) and CM (1577) find more adhesions than UN (1504). In missed objects, both OP ($\overline{A} = 0.61$ $\mu$m$^2$) and CM ($\overline{A} = 0.60$ $\mu$m$^2$) are significantly smaller than UN ($\overline{A} = 0.85$ $\mu$m$^2$). More adhesions are missed in UN (355) compared to OP (339) and CM (282) as well. Comparing the false positives, UN ($\overline{A} = 0.43$ $\mu$m$^2$) has the highest mean area $\overline{A}$ compared to OP ($\overline{A} = 0.36$ $\mu$m$^2$) and CM ($\overline{A} = 0.36$ $\mu$m$^2$). OP (745) and CM (231) have lower number of false positives compared to UN (1811) as well. Thus, for severely blurred images, optimization method shows improvement over unoptimized by finding more adhesions and having fewer false positives. Customization shows a greater improvement by having much fewer missed adhesions and false positives compared to other levels.

Thus FASensor offers several levels of desired user optimization in a variety of imaging conditions to derive and evaluate an accurate adhesion map from the input.

## Detection of stress fibers with FilamentSensor

The FilamentSensor, as integrated here in the FAFCK, is based on the version published by Eltzner et al in 2015 [4], adjusted to feature stack handling of image sequences and drastically reduced runtime as well as some additions for area calculation. The plugin featured in the FAFCK includes all components included in the stand-alone software, being a preprocessing, line sensor, and filament submenu. The workflow of the software as published before is included in S4 Fig.

During preprocessing, contrast and brightness can be adjusted for either individual pictures or a whole stack, if needed. These routines are based on ImageJ [21], which is included as an internal library and used wherever possible, as the ImageJ routines are fast and well tested. The main preprocessing step consists of a filter queue which the user can customize to their needs. This is necessary to prepare the original IF filament image for binarization, tackling the issue of crossing filaments that would otherwise be recognized as a network of interconnected, not crossing, filaments. On this image, the binarization is applied and filament objects extracted according to the flowchart shown in S4 Fig. This is done in parallel threads to improve runtime and subjected to several boundaries the user can determine including minimal and maximal length, maximal curvature, width, restriction to cell area mask, and more. This flexibility

allows for the program to be utilized for a wide variety of filament types. Lastly, the filament subsection allows to filter data for export. The FilamentSensor module offers a set of descriptors of the whole cell such as IDs, area, aspect ratio, length of axes, number of filaments, orientation, brightness, and for each individual filament such as xy position, length, curvature, width, and orientation. For each image file, the filament objects are assigned an individual identifier as done for the focal adhesion objects and a variety of export types are available with the option of superimposing filaments as required.

## Correlation of detected focal adhesions and actin filaments in FAFCK

As focal adhesions and actin filaments are linked structures, the FAFCK module offers correlation of detected focal adhesions from FASensor and filaments from FilamentSensor. The software's workflow is illustrated in Fig 7.

Using the file name of the original images or image stacks loaded, it is first checked whether input data for both adhesions and filaments exist and single sets are ignored. The focal adhesion objects detected from the input image showing paxillin are paired with the filaments derived from the input image depicting actin (see Fig 8A and 8B).

Both list of found objects are sorted by size and usually tasks start with the larges object. For correlation, one fiber object is taken and all focal adhesion objects are tried for correlation. To reduce computational efforts, only focal adhesions with long axis smaller than fiber total length are tried, assuming a focal adhesion can never be larger than the corresponding fiber. For the focal adhesion objects, the user decides whether the convex hull, fitted ellipse, or true pixels is used for verification purposes. This ellipse is calculated by setting the line between the two points with the greatest distance on the convex hull as long axis and the axis orthogonal to that and with the greatest length as short axis. Furthermore, the area can be artificially increased by increasing the neighborhood in which verification is done. Now, starting from the ends, for each point on the filament, a neighborhood rectangle is created and in the list of focal adhesions with main axis length below filament length, intersecting objects are searched.

The correlation can be done with condition of either validating all filaments that are attached to at least a single adhesion or only validating those with multiple adhesion structures along the filament. Thus, we can clearly categorize filaments by the number of adhesions associated. The data of adhesions by number of filaments associated can also be derived.

The output of the correlation routine consists of the identifier numbers of the respective objects and can consecutively be matched to the data output of the previous routines. Also, verified filaments will be highlighted in the fiber overlay and after verification the fiber data export will be expanded by a 'verification' column with booleans. As with the individual modules themselves, batch analysis of correlation for pairs of FA and SF images are possible as well.

The output is displayed in the Stress Fiber Overlay window in the main tab in FAFCK. The resulting paired filament and focal adhesion IDs are displayed in a table in the Focal Adhesion tab. The results can be exported as a simple overlay or a comprehensive color-coded map (Fig 8C), verifier tables, and grouped CSV files with details of adhesions and filaments by association with each other.

We used the FAFCK to correlate detected adhesions with filaments in the in-focus dataset (n = 17). We named filaments by adhesion association as MAAF- Multiple adhesion associated filament, SAAF—Single adhesion associated filament and NAAF—Not adhesion associated filament. There were 490 MAAFs, 588 SAAFs and 555 NAAFs, with MAAFs having significantly higher mean fiber length of 13.14 μm compared to SAAFs at 9.29 μm and NAAFs at 7.15 μm (Fig 8D). Thus, filaments attached to multiple adhesions are longer than those attached to only one adhesion or none. We also analyzed adhesions by number of fibers associated with them

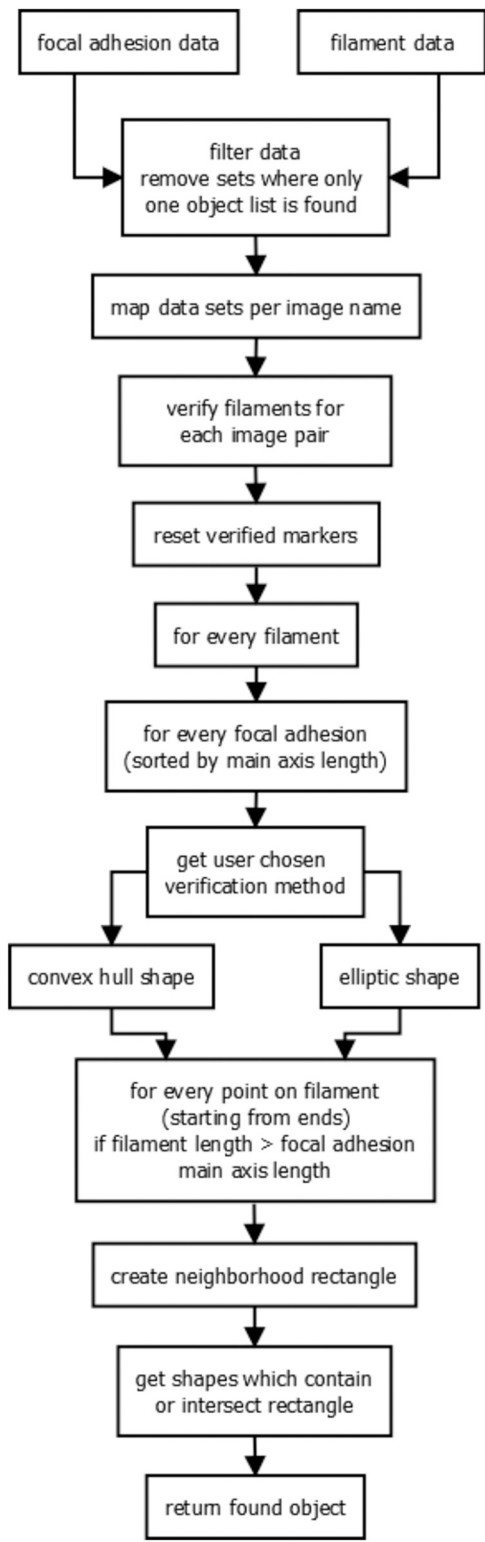

**Fig 7. Workflow for correlation in FAFCK.**

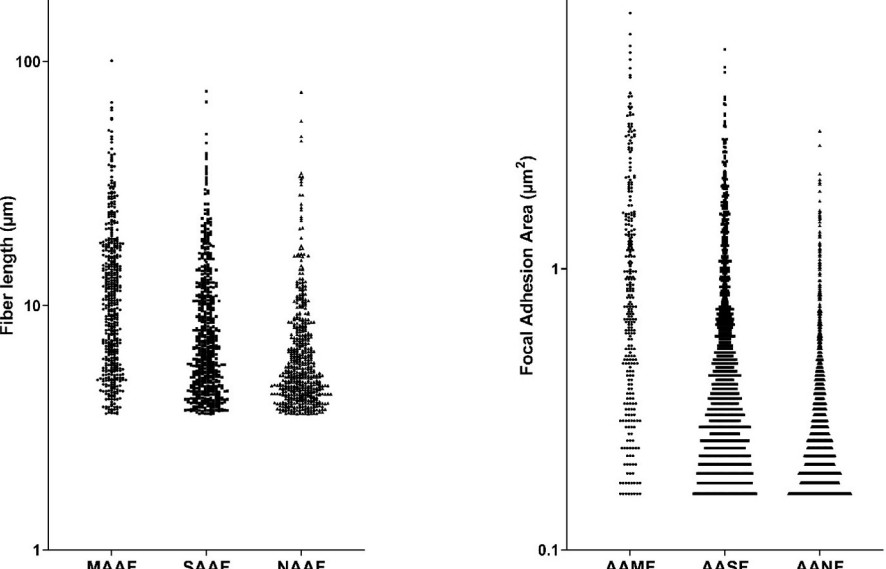

**Fig 8. Correlation of focal adhesions and actin filaments by FAFCK. A)** Input images of the ventral plane of a MRC5 cell (left) adhesions (paxillin) and (right) actin filaments (phalloidin). **B)** Map of numbered adhesion objects detected by FASensor (left), map of filaments detected by FilamentSensor (right). **C)** Color coded map of correlated adhesions and filaments categorized by association (legend in image). **D)** Aggregate graph of lengths of filaments categorized by adhesion association in all cells of the in-focus dataset. Y axis is in logarithmic scale. MAAF- Multiple adhesion associated filament, SAAF—Single adhesion associated filament and NAAF—Not adhesion associated filament. **E)** Aggregate graph of areas of adhesions categorized by filament association in all cells of in-focus dataset. Y axis is in logarithmic scale. AAMF—Adhesion associated with multiple fibers, AASF—Adhesion associated with single fiber and AANF—Adhesion associated with no fibers.

and grouped them as AAMF—Adhesion associated with multiple fibers, AASF—Adhesion associated with single fiber and AANF—Adhesion associated with no fibers. AAMFs were much fewer (291) than AASFs (1191) and AANFs (1227) and had significantly larger mean adhesion area ($\overline{A} = 1.21 \ \mu\text{m}^2$) compared to AASFs ($\overline{A} = 0.64 \ \mu\text{m}^2$) and AANFs ($\overline{A} = 0.34 \ \mu\text{m}^2$) (Fig 8E).

The correlation analysis provides a comprehensive picture of adhesion and filament association in cells, and can be used to streamline quantitative evaluation of the effective mechanical forces in the stress fiber / focal adhesion system.

## Discussion

Here we present the FAFCK that allows for fast, reliable, unbiased, and systematic detection of fibers and point-like structures and their cross-correlation in cells. While detection and analysis of both types of structures individually is useful, the cross-correlation module will be especially valuable and help to answer open questions on the coupled function of these force-transmitting features in cellular mechanosensing.

There are several notable advantages to our new tool. Importantly, it allows to identify groups of stress fibers associated with zero, one, or more than one focal adhesion. Such classification can be applied to functional differences of stress fibers in cells of specific morphologies. For example, in migrating cells, this allows for a quantification of the relative number and characteristics of transverse arcs (0 FA per filament), dorsal SFs (1 FA per filament), and ventral SFs ($\geq$2 FAs per filament) in large data sets. This analysis can also be applied to other types of actin organization in specialized cell types. The software package can also be used to quantify maximum intensity projections from 3D image sets, making it possible to quickly quantify such structures that would be otherwise difficult and time-consuming to analyze. Furthermore, individual application of filters and optimization allows for an optimal analysis of wide-field images and images with high blur and/or background noise.

There is always a certain degree of error or deviation in computational recognition methods (as false positives and false negatives) as well as bias in the user's native detection of cellular features. Our software package allows for the systematic, streamlined, and unbiased comparison of large data sets to achieve statistical relevance. Since we provide an option to customize output in each image, this also allows for more precise detection of SF types in smaller data subsets. We are currently developing the functionality of the FAFCK such that it would be useful for analysis of time lapse movies, where many frames need to be analyzed consecutively with same settings to quantify the dynamics of stress fibers and adhesions in cells to understand their dynamic organization and how they influence the mechanical coupling of cells and the matrix. While our original motivation for this project was the quantitative analysis of focal adhesion structures and their correlation with stress fibers, this tool can be also used for image analysis of other cellular structures from fluorescence microscopy images. This includes but is not limited to membrane organelles such as lysosomes or mitochondria, that can be detected and also tracked to quantify their cellular dynamics.

For optimal flexibility and potential comparative studies, we provide an import option of external data sets of filaments and FAs (source may be manual detection or from other software). This feature allows for comparison of the computational recognition with individual user perception of the biological reality and also allows for importing data from other image analysis platforms to be used for the correlation analysis. In the light of the continuous improvement of image recognition software in the field we specifically refrained from employing machine learning and big data algorithms to establish a solid classical analysis tool. That

said, the future development of FAFCK can surely benefit from big data and deep learning additions.

## Materials and methods

### Software availability

The FAFCK is available under the GNU Public License and can be used, modified and restributed freely without warranty given by the developers. A version of the software, sources, tutorial, installation notes, and example data can be either obtained by the data package associated with this paper (https://doi.org/10.5281/zenodo.5082933) or via our website (http://www.filament-sensor.de). The FAFCK has been tested on Windows and Linux. For Java 8 and below, the.jar file runs on click on windows. Running the FAFCK on Java 9 is not advised. For Java 10 and above, Java does not contain the JavaFX package, which has to be installed separately and the Path added. Also, we advise to use OpenJDK and OpenJFX which both have to be installed and the Path added on Windows and Linux. Detailed instructions how to do this can be found in the Readme in the data folder or on our website.

### Cell culture

MRC5 cells (human lung fibroblasts, ATCC$^{®}$ Cat# CCL-171™, RRID:CVCL_0440) were maintained in MEM media (Cat# 11095080, Thermo Fisher Scientific) supplemented with 10% fetal bovine serum, 100 μM penicillin and 0.1 mg/ml streptomycin in 5% $CO_2$ at 37˚C. Media was supplemented with 5 μg/ml Plasmocin (Cat# ant-mpp, InvivoGen) as a prophylactic against mycoplasma contamination.

### Fixation and immunostaining

MRC5 cells were seeded on glass coverslips (Cat# NC1129240, Fisher Scientific) that had been coated with 10 μg/ml fibronectin (Cat# FC010, EMD Millipore) for 1 hour. After 24 hours, cells were fixed with 4% paraformaldehyde prepared in CB (cytoskeletal buffer—150mM NaCl, 5mM $MgCl_2$, 5mM EGTA, 5mM glucose, 10mM MES), for 10 minutes at room temperature. They were washed with CB after fixation, permeabilized with 0.25% Triton in CB. Antibodies used are as follows: anti-paxillin mouse primary antibody (1:200, BD Biosciences Cat # 610051, RRID:AB_397463), Alexa Fluor 568 Phalloidin (1:300, Invitrogen, Cat# A12380) and Alexa Fluor 488 conjugated goat anti-mouse IgG secondary antibody (1:300, Thermo Fisher Scientific Cat# A-11001, RRID:AB_2534069). Coverslips were post-fixed for 10 min with 4% PFA in CB at room temperature. They were mounted with Vectashield Mounting Medium (Cat # H-1000–10, Vector Labs) on glass slides (Cat # 12–550-343, Fisher Scientific).

### Confocal microscopy

Immunostained samples were imaged using a laser scanning confocal microscope- Nikon A1R HD25 configured with a Ti2-E inverted microscope, with a 100× oil immersion objective (MRD01991, N.A. = 1.49). Three different pinhole settings were used to adjust the amount of out-of-focus light in the images- 1.2 AU (small, in-focus), 3 AU (intermediate, mild blur) and 4.7 AU (large, severe blur). Alexa Fluor 488 was excited with a laser of wavelength 488 nm and Alexa Fluor 568 with 561 nm, respectively.

### Input files from microscopy images

To ensure accurate analysis of the desired cell, we edited the IF images with multiple cells in the field of view by outlining the cell of interest, noting the background value and filling the

area outside the cell with the background. This edited image was taken as the input for the FASensor and FilamentSensor modules. In cases where simply cropping the image could isolate the cell of interest, we did so.

## Manual annotation by human expert

Images were manually marked for FAs by a human expert (17 images for in-focus set, 17 images for mild blur set, 19 images for severe blur set) and in addition the in-focus set was marked by a second independent human expert. FAs were marked using the freehand selection tool in Fiji [20]. The binary mask of marked adhesions were used as input in evaluation against the software's output.

## Bulk dataset evaluation analysis

To avoid detecting noise and artifacts, we set the lower limit of adhesion detection in the software to 10 pixels which corresponds to 0.144 μm$^2$ and upper limit at 1000 px which corresponds to 14.4 μm$^2$. The unoptimized routine across the imaging sets is as follows- For In-Focus dataset, Gauss filter (Sigma-1) and Laplace filter (1, 4 neighbor) with Intermodes thresholding at 55 was used. For Mild-Blur dataset, Gauss filter (Sigma-1) and Laplace filter (1.5, 8 neighbor) with Intermodes thresholding at 80 was used. For Severe-Blur dataset, Gauss filter (Sigma-1) and Laplace filter (3, 8 neighbor) with Intermodes thresholding at 80 was used. Further optimization and customization was done according to user discretion. Closing and Fill holes function was not used for bulk analysis adhesion detection. We used 1 percent minimum matching pixels for object matching in evaluation. Thicken lines function was not used in evaluation. Areas of found and missed adhesions were derived from the pixels column for the user's mask in the result table. Areas of false positive adhesions were derived from the pixels column of the software output in the result table. Pixel values from software results were converted to corresponding micron values using the scale of input image and plotted on graphs. Ordinary one-way ANOVA followed by Dunnett's multiple comparisons test or Tukey's multiple comparisons test was performed using GraphPad Prism (Ver 9.0.0 for Windows, GraphPad Software, San Diego, California USA, www.graphpad.com). For the in-focus set FA-filament correlation analysis, we used the optimization method where each cell had a custom optimal pre-processing filter setting in FASensor. For FilamentSensor, the default settings were used for all cells. For verification, we chose ellipse and a neighborhood of 1.

## Batch threshold determination with ThresholdFinder

The ThresholdFinder application is an additional software tool that we provide alongside the FAFCK. From a small amount of user-annotated masks, it determines best applicable thresholding algorithm and setting in the FAFCK software. From the input of original images and binary annotations, the software uses the mask to determine desired regions of the image and feeds this into all thresholding algorithms. To determine found, false positive and false negative rates, the images are processed whole with the selected algorithm. The value that the respective algorithm would chose without mask input is given, too.

## Supporting information

**S1 Fig. Evaluation of adhesions marked by user expert in Fiji vs those detected by the software module. A**) FASensor input of IF adhesion image showing the ventral plane of a MRC5 cell immunostained with paxillin. **B**) Binary mask of adhesion ROIs marked by user expert **C**) (Left-Right) Output adhesion objects detected by FASensor from input of A, Objectwise

evaluation map of mask vs output, Pixelwise evaluation map of mask vs output. (in all evaluation images, found- blue, missed-yellow, false positive-red). **D**) Objectwise evaluation results tabulated. **E**) Pixelwise evaluation results tabulated.
(PNG)

**S2 Fig. Using an optimal input routine increases the similarity of output with user mask.**
**A**) Grayscale image of adhesions, ventral plane of MRC5 cell immunostained for paxillin. **B**) Binary mask of adhesion ROIs manually marked by user expert from A through Fiji software. **C**) Un-optimized output vs user mask comparison (Left) FASensor output (Right) Objectwise evaluation map (found- blue, missed-yellow, false positive-red) **D**) Optimized output vs user mask comparison (Left) FASensor output (Right) Objectwise evaluation map (found- blue, missed-yellow, false positive-red) **E**) Table comparing results between un-optimized and optimized evaluations.
(PNG)

**S3 Fig. False positives and missed adhesions are much smaller than found adhesions.**
Graphs show pooled adhesion objects for found, missed and false positive (FP) categories in a set. Y axis is adhesion area in $\mu m^2$ on a logarithmic scale. Left column shows un-optimized setting graphs, middle column shows optimized setting graphs and right column shows customized setting graphs. **A**) In focus set (Left) Found n = 1451 $\overline{A} = 1.73$ $\mu m^2$, Missed n = 660 $\overline{A} = 0.86$ $\mu m^2$ and FP n = 497 $\overline{A} = 0.43$ $\mu m^2$; (Middle) Found n = 1843 $\overline{A} = 1.57$ $\mu m^2$, Missed n = 268 $\overline{A} = 0.71$ $\mu m^2$ and FP n = 465 $\overline{A} = 0.25$ $\mu m^2$; (Right) Found n = 1802 $\overline{A} = 1.59$ $\mu m^2$, Missed n = 309 $\overline{A} = 0.68$ $\mu m^2$ and FP n = 296 $\overline{A} = 0.33$ $\mu m^2$; **B**) Mild-blur set (Left) Found n = 1522 $\overline{A} = 1.64$ $\mu m^2$, Missed n = 579 $\overline{A} = 0.83$ $\mu m^2$ and FP n = 791 $\overline{A} = 0.42$ $\mu m^2$; (Middle) Found n = 1740 $\overline{A} = 1.57$ $\mu m^2$, Missed n = 361 $\overline{A} = 0.66$ $\mu m^2$ and FP n = 560 $\overline{A} = 0.31$ $\mu m^2$; (Found n = 1700 $\overline{A} = 1.60$ $\mu m^2$, Missed n = 401 $\overline{A} = 0.64$ $\mu m^2$ and FP n = 188 $\overline{A} = 0.43$ $\mu m^2$; **C**) Severe-blur set (Left) Found n = 1504 $\overline{A} = 1.61$ $\mu m^2$, Missed n = 355 $\overline{A} = 0.85$ $\mu m^2$ and FP n = 1811 $\overline{A} = 0.43$ $\mu m^2$; (Middle) Found n = 1520 $\overline{A} = 1.66$ $\mu m^2$, Missed n = 339 $\overline{A} = 0.61$ $\mu m^2$ and FP n = 745 $\overline{A} = 0.36$ $\mu m^2$; (Right) Found n = 1577 $\overline{A} = 1.62$ $\mu m^2$, Missed n = 282 $\overline{A} = 0.60$ $\mu m^2$ and FP n = 231 $\overline{A} = 0.36$ $\mu m^2$.
(PNG)

**S4 Fig. Comparison with annotations from two human experts. A**) FASensor input of IF adhesion image showing the ventral plane of a MRC5 cell immunostained with paxillin. **B**) Binary mask of adhesion ROIs marked by user expert 1. **C**) Binary mask of adhesion ROIs marked by user expert 2. **D**) Evaluation output for pixel-wise and object-wise comparison of human expert annotations compared to FASensor output. **E**) Similarity coefficient of human expert annotated masks, human expert 1 ($\overline{\mathcal{SC}} = 4.53$), human expert 2 ($\overline{\mathcal{SC}} = 4.63$).
(PNG)

**S5 Fig. Workflow for FilamentSensor.**
(PNG)

## Acknowledgments

We thank Hamida Ahmed for technical assistance.

## Author Contributions

**Conceptualization:** Lara Hauke, Irina Kaverina, Florian Rehfeldt.

**Data curation:** Lara Hauke, Shwetha Narasimhan.

**Formal analysis:** Shwetha Narasimhan.

**Funding acquisition:** Irina Kaverina, Florian Rehfeldt.

**Investigation:** Lara Hauke, Shwetha Narasimhan, Florian Rehfeldt.

**Methodology:** Lara Hauke, Andreas Primeßnig.

**Project administration:** Florian Rehfeldt.

**Resources:** Irina Kaverina, Florian Rehfeldt.

**Software:** Lara Hauke, Andreas Primeßnig.

**Supervision:** Irina Kaverina, Florian Rehfeldt.

**Visualization:** Lara Hauke, Shwetha Narasimhan.

**Writing – original draft:** Lara Hauke, Shwetha Narasimhan, Florian Rehfeldt.

**Writing – review & editing:** Lara Hauke, Shwetha Narasimhan, Irina Kaverina, Florian Rehfeldt.

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
