## [Decision Letter · Decision Letter 0]

10 May 2021

PONE-D-21-10621

A Focal Adhesion Filament Cross-correlation Kit for fast, automated segmentation and correlation of focal adhesions and actin stress fibers in cells

PLOS ONE

Dear Dr. Rehfeldt,

Thank you for submitting your manuscript to PLOS ONE. After careful consideration, we feel that it has merit but does not fully meet PLOS ONE’s publication criteria as it currently stands. Therefore, we invite you to submit a revised version of the manuscript that addresses the points raised during the review process.

Based on recommendation of both Reviewers', the authors should demonstrate an application of this software for the analysis of time-lapse images potentially through tracking the stress fibers and focal adhesions over time. The authors should provide additional details on the algorithm used to optimize the image processing to match with a user input binary adhesion map. The authors should include additional details regarding where to download the software and what operating systems it has been tested on. The authors should also consider to include the figure demonstrating a software interface for each corresponding operation.

We look forward to receiving your revised manuscript.

Kind regards,

Yulia Komarova

Academic Editor

PLOS ONE

Journal Requirements:

3. Thank you for stating the following in the Acknowledgments and Funding Section of your manuscript:

[This publication was funded by the German Research Foundation (DFG) and the University of Bayreuth in the funding programme Open Access Publishing.]

 [I.K. ; R35-GM127098  ; The National Institute of General Medical Sciences (NIGMS) of the National Institutes of Health (NIH);  https://www.nigms.nih.gov/

I.K. ; R01-DK106228  ; The National Institute of Diabetes and Digestive and Kidney Diseases (NIDDK) of the National Institutes of Health (NIH);  https://www.niddk.nih.gov/

S.N. ; 18PRE33990479  ;  https://www.heart.org/  ; the American Heart Association (AHA) ;  https://www.heart.org/

F.R. SFB 755 - B08, Deutsche Forschungsgemeinschaft (DFG), www.dfg.de

F.R. SFB 937 - A13, Deutsche Forschungsgemeinschaft (DFG), www.dfg.de

The funders had no role in study design, data collection and analysis, decision to publish, or preparation of the manuscript.]

Reviewers' comments:

Reviewer's Responses to Questions

**Comments to the Author**

1. Is the manuscript technically sound, and do the data support the conclusions?

Reviewer #1: Partly

Reviewer #2: Yes

2. Has the statistical analysis been performed appropriately and rigorously? 

Reviewer #1: Yes

Reviewer #2: Yes

3. Have the authors made all data underlying the findings in their manuscript fully available?

Reviewer #1: Yes

Reviewer #2: Yes

4. Is the manuscript presented in an intelligible fashion and written in standard English?

Reviewer #1: Yes

Reviewer #2: Yes

5. Review Comments to the Author

Reviewer #1: This paper describes a new software package dedicated to quantifying the relationship between focal adhesions and stress fibers. Both of these structures are well studied and software to understanding how they interact from a quantitative perspective would be a nice addition. Overall, I like the concept for the paper and quite a bit of effort has been put in to test the capabilities of the software. I’m a bit concerned that the paper was submitted without any means to download or any instructions for how to use the software, but I think the authors can fix these problems.

Major Issues:

I can’t get your software to run on my computer. After downloading the jar file from zenodo, I ran:

~/Downloads$ java -jar GUIFocalAdhesionOnly.jar

Error: Could not find or load main class fa.view.MainWindow

Caused by: java.lang.NoClassDefFoundError: javafx/application/Application

This might be because I run Linux on my desktop. Since I didn’t see any information in your paper about what operating systems you’ve tested, I wanted to bring this up as a potential issue. I could be making fundamental error as I don’t often run java programs like this, but I couldn’t find a README file either. Is this supposed to be a plugin for FIJI/ImageJ? If the other reviewers/editor didn’t have any problem using the software then this can be disregarded.

A section concerning software availability needs to be added to the materials and methods. This can be a short section, but it should detail where to download the software and on what operating systems it has been tested. Since the code is released under the GPL the source code should also be made available, preferable through github or a similar source code sharing website. The zenodo link provided should be integrated into this section.

The description of the software interface on Lines 85-98 is confusing without a figure showing the software interface. I understand that you might not want to add a figure, but I find it strange to be reading a description of a software interface without actually having a figure to reference.

The description of optimizing the processing algorithm to match with a user input binary adhesion map from Lines 153-186 is really interesting, but I can’t tell from the description what aspects of the algorithm were modified to optimize the result. Is this an automated process built into the software? In other words, can a user provide their ideal segmentation results and get back an algorithm parameter configuration to match up with the manual segmentation? This seems to be sort of covered in the materials and methods (Batch threshold determination... ), but this section also doesn’t detail what parameters are modified.

My conclusion from the of the “FASensor output performance with varying imaging conditions

and levels of optimization” section is that your software should not be used without a fair bit of effort being put in to customize the settings for every single image analyzed. I understand that settings need to be checked for a given set of images, but how bad is it to use the same settings for a set of images all gathered at the same time? Otherwise, I think this entire section comes down to the quality of the ground truth results. Since it seems like a single person was responsible for gathering these results, I’d like to see what the similarity coefficient is for independent manual segmentation results. Maybe you don’t need to customize the settings at all because independent experts can’t really decide what an FAs is in your sample images anyway.

The conclusion mentions analysis of dynamics (Line 389 and 397), but I don’t see any evidence that the tool can handle dynamic or time-lapse analysis. I would expect that this would require tracking the stress fibers and FAs through time, as opposed to treating every image individually. These sections in the conclusion should be rewritten to indicate that the software could be extended to have this functionality.

Minor Issues:

Line 2: “cellular endoskeleton”: I’ve never heard the cytoskeleton called an endoskeleton, but I don’t suppose it’s wrong.

Line 16: “order parameter S”: I’m not really sure what this means, is this a reference to a property you are going to calculate later in the paper?

Line 130: “which is accurate with user’s expert perception of the IF image.” Maybe this should be a new sentence? Something like “After the reprocessing the user is able to confirm that the newly joined adhesion matches with expectations.”

Line 134: “between the user expert’s routine” Should this be the “user’s usual routine”?

Line 145: “with user mask” Should this be “with the user’s mask”?

Line 298: “This serves as the ’artificial retina’” I really don’t know how the beginning of this sentence is related to the end. What is an artificial retina? Is the CMOS camera you used to gather the images an “artificial retina”? How does this help segment crossing filaments?

Line 321: “Starting from the ends, for every point in a filament, every focal adhesion in the

specified neighborhood is checked with the requirement that the focal adhesion main

axis has to be longer than the filament length.” Does this sentence mean that the first filter tosses out any association between a FA with a shorter main axis as compared to the filament length? My intuition would be that nearly every filament will be longer than the associated FA, seeing as a large percentage of the filaments cover a substantial portion of the cell.

Line 368: “help to answer many burning questions” I would strongly encourage the authors to consider not using the adjective burning here.

Line 374: “an quantification”, I think this should be “a quantification”

Line 378: “functionality allows to recognize” should be “functionality allows the recognition of”

Line 378-381: “This software functionality allows to recognize different types of actin bundles in

2-dimensional images, such as maximum intensity projections of confocal stacks, which

significantly quickens the quantification (in comparison with the necessity to detect

structures in three-dimensional space, for example, dorsal stress fibers ).”

This sentence is really difficult to follow, maybe something like this? “The software package can also be used to quantify maximum intensity projections from 3D image sets, making it possible to quickly quantify these difficult to measure structures.”

Reviewer #2: The authors reported an extended software module from the existing Focal Adhesion (FA) Sensor. This software provides an automated analysis of focal adhesion and Stress fiber. This software is multi-functional that fit the proposed purpose very well, also considered majority concerns of common images processing. The instruction is clear instruction and easy to follow. This software will benefit wide range of biological research that associated with Focal Adhesion and stress fiber. The manuscript is fairly completed in my opinion. Still, I have the following comments and would like to see the authors respond:

Major:

-Demonstrating biological relevance will provide a great substantiation of application of the software. Have the authors attempted to correlate the image analysis with migrating cell as the author hypothesized in discussion section? Or correlate the image analysis with experimentally measured mechanical properties of the MRC5 cell since FA and stress fiber play an important role for mechanosensing. The current imaging was only done in one culturing condition: MRC5 cells seeded on glass coverslips, which is a very rigid substrate.

-Screen capture for the panels of the software for each corresponding operation will be helpful. Right now, the operation steps are all described solely by words, so the reader may not be able to fully appreciate the software.

-Supplement Fig 3 seems to be an important evaluation of the software, I suggest to move it to main figure and combine with figure 4.

Minor:

-Line 53 and 55: it is recommended to state what is “other methods” and “other tools” as the references cited are clear.

-Fig 3C is presented before Fig 3A and B. I suggest to swap the order to match the presentation flow.

-In general, the quality of the figures should be improved. From my end, most of the figures are pixelized, that include fluorescent images and the data figures. Especially for the legend in Fig 7C, that is un-readable.

6. PLOS authors have the option to publish the peer review history of their article (what does this mean?). If published, this will include your full peer review and any attached files.

Reviewer #1: No

Reviewer #2: No

---

## [Author Response · Author response to Decision Letter 0]

9 Jul 2021

We are grateful for the comments of the reviewers and answered them point-by-point in the response letter.

---

## [Decision Letter · Decision Letter 1]

30 Jul 2021

PONE-D-21-10621R1

A Focal Adhesion Filament Cross-correlation Kit for fast, automated segmentation and correlation of focal adhesions and actin stress fibers in cells

PLOS ONE

Dear Dr. Rehfeldt,

Thank you for submitting your manuscript to PLOS ONE. After careful consideration, we feel that it has merit but does not fully meet PLOS ONE’s publication criteria as it currently stands. Therefore, we invite you to submit a revised version of the manuscript that addresses the points raised during the review process.

1. The authors should include figure 2 for review as it was missing in R1 version of the manuscript.

2. The authors might consider including some additional information/steps to the installation instructions in order to help with an application of this tool by the research community

We look forward to receiving your revised manuscript.

Kind regards,

Yulia Komarova

Academic Editor

PLOS ONE

Journal Requirements:

Additional Editor Comments (if provided):

Reviewers' comments:

Reviewer's Responses to Questions

**Comments to the Author**

1. If the authors have adequately addressed your comments raised in a previous round of review and you feel that this manuscript is now acceptable for publication, you may indicate that here to bypass the “Comments to the Author” section, enter your conflict of interest statement in the “Confidential to Editor” section, and submit your "Accept" recommendation.

Reviewer #1: All comments have been addressed

Reviewer #2: (No Response)

2. Is the manuscript technically sound, and do the data support the conclusions?

Reviewer #1: Yes

Reviewer #2: Yes

3. Has the statistical analysis been performed appropriately and rigorously? 

Reviewer #1: Yes

Reviewer #2: Yes

4. Have the authors made all data underlying the findings in their manuscript fully available?

Reviewer #1: Yes

Reviewer #2: Yes

5. Is the manuscript presented in an intelligible fashion and written in standard English?

Reviewer #1: Yes

Reviewer #2: Yes

6. Review Comments to the Author

Reviewer #1: All of my comments have been addressed and the additional figure makes that section of the paper much easier to follow. I suggest the paper be published without further revision.

With the new instructions I was able to get the software to run on my computer (Ubuntu 21.04). I had to install two packages:

sudo apt install default-jre

sudo apt install openjfx

The command you included in your installation info didn’t quite work for me, but a few minutes of googling led me to this command that worked for starting the software:

java --module-path /usr/share/openjfx/lib --add-modules=javafx.base,javafx.controls,javafx.fxml,javafx.graphics,javafx.media,javafx.swing,javafx.web -jar GUIFocalAdhesionOnly.jar

This might be something unique about how Ubuntu handles library installations, but I also don’t know the Java world very well. Maybe this extra info might be nice to add to the installation instructions, otherwise, thanks for addressing all my concerns.

Reviewer #2: Major issue of the revised manuscript:

I couldn’t see Figure 2 in the revised manuscript. I assume this is just uploading issue of the figure as the authors have clear context related Figure 2 in the main text.

Comment on the revised manuscript:

I suggested the author to demonstrate the biological relevance of the software, but the author would like to have this manuscript to be focus on introducing FAFCK as a novel tool that potential beneficial to the community. I agree that reporting the software alone with substantial elaboration, as the authors did in this manuscript, can be an independent, technical manuscript. Therefore, I am not pushing further for the additional biological experiments.

The authors provided substantial supporting materials as a tutorial of the software for readers, I am satisfied with the authors response.

The authors also addressed all of my other concerns.

7. PLOS authors have the option to publish the peer review history of their article (what does this mean?). If published, this will include your full peer review and any attached files.

Reviewer #1: No

Reviewer #2: No

---

## [Author Response · Author response to Decision Letter 1]

5 Aug 2021

We are happy that our revisions were satisfying the referees concerns and apologize for the omission of Figure 2. We added this Figure (the same as in the original submission) to the list of Figures.

We are also grateful for the comment of reviewer one concerning the installation of the software on a Linux system running Ubuntu 21.04. We added the following information as comment to the repository:

We got feedback that a system running Ubuntu 21.04 needed a slightly modified command line:

java --module-path /usr/share/openjfx/lib –add modules=javafx.base,javafx.controls,javafx.fxml,javafx.graphics,javafx.media,javafx.swing,javafx.web -jar GUIFocalAdhesionOnly.jar

We hope that after these additions the manuscript can be published in PLoS One.

---

## [Decision Letter · Decision Letter 2]

24 Aug 2021

A Focal Adhesion Filament Cross-correlation Kit for fast, automated segmentation and correlation of focal adhesions and actin stress fibers in cells

PONE-D-21-10621R2

Dear Dr. Rehfeldt,

We’re pleased to inform you that your manuscript has been judged scientifically suitable for publication and will be formally accepted for publication once it meets all outstanding technical requirements.

Kind regards,

Yulia Komarova

Academic Editor

PLOS ONE

Additional Editor Comments (optional):

Reviewers' comments:

Reviewer's Responses to Questions

**Comments to the Author**

1. If the authors have adequately addressed your comments raised in a previous round of review and you feel that this manuscript is now acceptable for publication, you may indicate that here to bypass the “Comments to the Author” section, enter your conflict of interest statement in the “Confidential to Editor” section, and submit your "Accept" recommendation.

Reviewer #2: All comments have been addressed

2. Is the manuscript technically sound, and do the data support the conclusions?

Reviewer #2: Yes

3. Has the statistical analysis been performed appropriately and rigorously? 

Reviewer #2: Yes

4. Have the authors made all data underlying the findings in their manuscript fully available?

Reviewer #2: Yes

5. Is the manuscript presented in an intelligible fashion and written in standard English?

Reviewer #2: Yes

6. Review Comments to the Author

Reviewer #2: (No Response)

7. PLOS authors have the option to publish the peer review history of their article (what does this mean?). If published, this will include your full peer review and any attached files.

Reviewer #2: No

---

## [Editor Report · Acceptance letter]

3 Sep 2021

PONE-D-21-10621R2 

A Focal Adhesion Filament Cross-correlation Kit for fast, automated segmentation and correlation of focal adhesions and actin stress fibers in cells. 

Dear Dr. Rehfeldt:

I'm pleased to inform you that your manuscript has been deemed suitable for publication in PLOS ONE. Congratulations! Your manuscript is now with our production department. 

Kind regards, 

on behalf of

Dr. Yulia Komarova 

Academic Editor

PLOS ONE